# Comparison of Long-Term Albedo Products against Spatially Representative Stations over Snow

Ruben Urraca [1,*] , Christian Lanconelli [1,2] , Fabrizio Cappucci [1,2] and Nadine Gobron [1]

1 EuropeanCommission, Joint Research Centre, Via Fermi 2749, I-21027 Ispra, Italy
2 UniSystems SA, Rue du Puits Romain 29, L-8070 Bertrange, Luxembourg
* Correspondence: ruben.urraca-valle@ec.europa.eu

**Abstract:** Multiple satellite products are available to monitor the spatiotemporal dynamics of surface albedo. They are extensively assessed over snow-free surfaces but less over snow. However, snow albedo is critical for climate monitoring applications, so a better understating of the accuracy of these products over snow is needed. This work analyzes long-term (+20 years) products (MCD43C3 v6/v6.1, GLASS-AVHRR, C3S v1/v2) by comparing them against the 11 most spatially representative stations from FLUXNET and BSRN during the snow-free and snow-covered season. Our goal is to understand how the performance of these products is affected by different snow cover conditions to use this information in an upcoming product inter-comparison that extends the analysis spatially and temporally. MCD43C3 has the smallest bias during the snow season ($-0.017$), and more importantly, the most stable bias with different snow cover conditions. Both v6 and v6.1 have similar performance, with v6.1 just increasing slightly the coverage at high latitudes. On the contrary, the quality of both GLASS-AVHRR and C3S-v1/v2 albedo decreases over snow. Particularly, the bias of both products varies strongly with the snow cover conditions, underestimating albedo over snow and overestimating snow-free albedo. GLASS bias strongly increases during the melting season, which is most likely due to an artificially extended snow season. C3S-v2 has the largest negative bias overall over snow during both the AVHRR ($-0.141$) and SPOT/VGT ($-0.134$) period. In addition, despite the improvements from v1 to v2, C3S-v2 still is not consistent enough during the transition from AVHRR to SPOT/VGT.

**Keywords:** albedo; snow; satellite products; FLUXNET; MODIS; BSRN

## 1. Introduction

Surface albedo is defined as the ratio of the hemispherical irradiance reflected from the surface to the incoming irradiance upon the surface [1]. It is one of the biosphere Essential Climate Variables (ECV) of the Global Climate Observing System (GCOS) [2] due to its key role in the Earth radiation budget. Surface albedo is highly variable both spatially and temporally. Temporal changes are driven by seasonal changes in vegetation and snow, fires, anthropogenic land cover changes, and illumination changes caused by annual and diurnal solar cycles. Measuring albedo over snow is particularly important for climate change monitoring. The global snow cover retreat observed during the last few years [3,4] is triggering a positive climate that reinforces the snow cover loss and the effects of climate change. The Intergovernmental Panel on Climate Change (IPCC) Sixth Assessment Report (AR6) [5] identified surface albedo as currently the third-largest positive climate feedback with an estimated value of +0.35 (+0.1 to +0.6) W/m$^2$/°C.

Satellite products provide the best way to monitor the spatiotemporal changes of surface albedo. Multiple satellite-based products are available at a global scale. Copernicus Climate Change Service (C3S) produces a multi-sensor product (v1 and v2, 1981–present) with AVHRR (4 km), SPOT/VGT (1/112°) and PROBA-V (1/112°) observations [6] as well as a product (v3, 2018–present) based on Sentinel-3/OLCI (300 m). Several albedo

products exploit NASA's optical senors: MCD43A3 (500 m) and MCD43C3 (0.05°) from Terra-Aqua/MODIS (2000–present) [7], MIL2ASLS (1.1 km) and MIL3MLSN (0.5°) from Terra/MISR (2000–present) [8,9], and VNP43IA3N (500 m) and VNP43MA3N (1 km) from SuomiNPP/VIIRS (2015-present) [10]. EUMETSAT Satellite Application Facilities (SAFs) offer ETAL (2015–present, 0.01°) within the Land Surface Analysis SAF (LSA-SAF), and CLARA (1981–present, 0.25°) within the Climate Monitoring SAF (CM-SAF) [11]. GlobAlbedo (1998–2011, 1 km) was produced by ESA from MERIS and SPOT/VGT sensors [12]. GlobAlbedo was enhanced in the QA4ECV project including AVHRR, PROBA-V and MVIRI/SEVIRI data [13]. The GLASS suite offers two products derived from MODIS (2000–present, 1 km and 0.05°) and one from NOAA/AVHRR (1981-present, 0.05°) [14]. Surface albedo can be also derived from high-resolution optical images from Landsat8/OLI (30 m) [15] or Sentinel-2/MSI (10–60 m) [16], but high-resolution operational products from those sensors are not available yet.

Several studies have evaluated the accuracy of albedo products through comparison against in situ measurements. The main challenge faced by these studies is the low spatial representativeness of in situ measurements due to the high spatial variability of albedo and the low footprint of in situ sensors. This issue is addressed using high-resolution products either to discard stations with low representativeness [17,18] or to up-scale the in situ measurements to the satellite resolution [19,20]. However, both approaches introduce uncertainty in the comparisons, particularly in regions with high spatiotemporal albedo changes.

Most studies assess albedo products over snow-free surfaces [17–21]. However, satellite products typically have lower accuracy over snow due to the different anisotropic reflectance of different types of snow and the typically high solar zenith angles (SZA) of satellite observations over snow in polar regions. Satellite products do not have the adequate resolution to gather the high spatiotemporal variability of snow cover in some regions. The snow-masking algorithms used by each product introduce additional uncertainties that can be particularly relevant during the snow onset and melting periods. Despite the limitations of albedo products over snow, and its importance for climate monitoring, validations over snow are less frequent [22–24], which is partly due to the lower number of stations in snow-covered regions and the higher number of gaps in the products over snow. A better understanding of the performance of albedo products over snow is needed.

Our ultimate goal is to evaluate the fitness of long-term (+20 years) products to monitor global albedo changes over snow. A global assessment of long-term albedo products can only be made by inter-comparing products due to the scarcity of long-term albedo stations, particularly in snow-covered regions. In this study, as a first step, we evaluate how the performance of each product changes with snow cover conditions (from fully covered to snow-free) by comparing them against in situ observations. We consider all satellite products covering 20+ years with a resolution of 0.05° × 0.05° or higher: GLASS-AVHRR 4.2 (1981–present), C3S v1/v2 multi-sensor (1981–2020), and MCD43C3 v6/v6.1 (2000–present). First, we analyze the spatial representativeness of all FLUXNET [25] and Baseline Surface Radiation Network (BSRN) [26] stations measuring albedo over snow. The 11 most spatially representative stations are then used to evaluate the accuracy of the products from 2000/02 to 2014/05, which is the period when most products and stations are simultaneously available.

## 2. Materials and Methods

### 2.1. Satellite Products

#### 2.1.1. GLASS-AVHRR

The Global Land Surface Satellite (GLASS) suite, which is produced by the Beijing Normal University, offers albedo products derived from MODIS (500 × 500 m & 0.05 × 0.05°) and AVHRR (0.05 × 0.05°) sensors (Table 1). All of them provide shortwave (300–3000 nm), visible (300–700 nm) and NIR (700–3000 nm) black- and white-sky albedo every 8 days. Surface albedo is derived using the direct-estimation method, which compared to multi-

angular methods generates one broadband albedo estimate from each Top-of-Atmosphere (TOA) reflectance measurement [27]. The relationship between TOA measurements and broadband surface albedo is characterized by a training dataset obtained by coupling a POLDER-based bidirectional reflectance distribution function (BRDF) database with 6S (second simulation of a satellite signal in the solar spectrum) atmospheric radiative transfer simulations [28]. Then, a linear regression model is fit for each angular bin and land use (vegetation, soil, and snow/ice). Surface broadband albedo is obtained by applying the fitted models to each satellite observation. The product is smoothed and gap filled with a spatiotemporal filtering algorithm [29].

The POLDER BRDF was interpolated with a linear kernel model to increase the number of observations between SZA 20° to 60°. The interpolation was made with the linear kernel-driven model proposed by Wanner et al. [30] modified to account for the forward-scattering effects of snow/ice [31]. The POLDER BRDF was screened based on empirical thresholds [27] to remove snow observations with high reflectance variability during the collection period (one month). The POLDER BRDF database was divided into 3 classes to train the angular bin regression model: vegetation, soil, and snow/ice. Classes were defined based on the Normalized Difference Vegetation Index (NDVI) and TOA reflectance in band 490 nm ($r_{490}$). The snow/ice class is composed by two subclasses to smooth the transition periods: pure snow ($r_{490} > 0.4$) and intermediate class B ($0.25 < r_{490} < 0.4$). The class mask is not available in the final product. Compared to products based on the multi-date models (MCD43, C3S albedo), the direct estimation method should capture better the rapidly changing albedo during snowfall and snowmelt seasons, as each value averages the direct estimations made from all observations available during the 8-day window [32].

This study evaluates the GLASS-AVHRR v4 ($0.05 \times 0.05°$), which was released in 2019 and includes an updated snow/ice BRDF training dataset and a new water surface BRDF model for mixed pixels of water/sea ice [31].

**Table 1.** Summary of the satellite products evaluated.

| Product | Sensor | Spatial Coverage | Temporal Coverage | Spatial Resolution | Temporal Coverage | Method |
|---|---|---|---|---|---|---|
| GLASS 4.2 | NOAA/AVHRR | global | 1982–2019 | $0.05 \times 0.05°$ | 8 d (8 d) | direct estimation |
| MCD43C3 6 & 6.1 | Terra-Aqua/MODIS | global | 2000/02–2020 | $0.05 \times 0.05°$ | 1 d (16 d) | BRDF inversion RossThick-LiSparseReciprocal |
| C3S albedo 1&2 | NOAA/AVHRR SPOT/VGT PROBA-V | 80N-60S | 1982/01–2005/12 1998/04–2014/05 2014/01–2020/06 | $1/30 \times 1/30°$ $1/112 \times 1/112°$ $1/112 \times 1/112°$ | 10 d (20 d) | BRDF inversion RossThick-LiSparseReciprocal |

### 2.1.2. MCD43C3

MCD43 products are developed by NASA from the MODIS sensor on-board Terra and Aqua satellites. Different BRDF and albedo products are available with a maximum resolution of 500 m. In this study, we evaluate the MCD43C3 Albedo Product (MODIS Albedo Daily L3 Global 0.05° CMG) which provides spectral (MODIS bands 1 to 7) and broadband (visible 300–700 nm, NIR 700–5000 nm and shortwave 300–5000 nm) white- and black-sky albedo [7]. MCD43C3 is derived from MCD43C1 and re-projected to a climate modeling grid of $0.05 \times 0.05°$. MCD43 BRDF/albedo is produced daily using 16 days of MODIS observations weighted accounting for their quality, spatial coverage, and the temporal distance from the day of interest. A full inversion of the RossThick-LiSparseReciprocal BRDF [33] is made when at least seven observations are available. Otherwise, a database of archetypal BRDF parameters is used to supplement the observational data and perform a lower quality magnitude inversion.

Snow-covered and non-snow observations are processed separately. If most of the 16-day window is snow-covered, snow-free observations are discarded, and vice versa. The MODIS-derived archetypal database contains both snow and snow-free representations.

The model, however, presents some limitations over snow. First, the use of a 16-day window assumes that the geophysical system under analysis does not experience significant changes during this period [34]. This assumption may not hold during windows with highly temporally changing snow cover conditions. Second, the Ross-Thick Li-Sparse BRDF was originally designed for the simplified scenarios of continuous and discrete vegetation canopies and has been mainly used to model the reflectance anisotropy of soil–vegetation systems, which tends to emphasize strong backward-scattering effects. Thus, the model has limitations to represent snow-scattering properties that in theory also present strong forward-scattering effects, particularly at larger viewing and solar geometries [35].

MCD43 is a clear sky product, so the algorithm is not designed to be specifically robust against conditions of increased haziness and $SZA > 75°$. The final product includes a quality flag and the percent of snow at each pixel. We evaluate both v6 and the new v6.1, which includes changes to the response-versus-scan angle approach that affects reflectance bands for Aqua-Terra/MODIS, corrections to adjust for the optical crosstalk in Terra/MODIS infrared bands, and corrections to the Terra/MODIS forward look-up table update for the period 2012–2017.

### 2.1.3. C3S Surface Albedo

The Copernicus Climate Change Service (C3S) produces a 10-daily multi-sensor surface albedo product that spans from 1981 to 2020 using NOAA-[7, 9, 11, 14, 16, 17]/AVHRR-[2, 3], SPOT-[4, 5]/VGT-[1, 2], and PROBA-V observations [36]. The parameters provided are 10-day spectral and broadband white and black sky albedo with a spatial resolution of $1/30°\sim4$ km (AVHRR) and $1/112°\sim1$ km (SPOT/VGT and PROBA-V). Broadband albedo is available for the shortwave (300–4000 nm), visible (400–700 nm) and NIR (700–4000) ranges.

The processing algorithm was developed by Meteo-France based on MSG/SEVIRI and Metop/AVHRR within EUMETSAT LSA-SAF and later adapted to these sensors by C3S and Copernicus Global Land Service (CGLS). TOA satellite measurements are atmospherically corrected with the Simplified Model for Atmospheric Correction (SMAC) [37] and harmonized to the spectral bands of SPOT/VGT-2. Bands 1 and 2 are used for AVHRR/2, and bands 1, 2 and 3A are used for AVHRR/3. The RossThick-LiSparse-Reciprocal BRDF model is inverted every 10 days (10th, 20th, 30th of each month) using a compositing window of 20 days. The previous BRDF inversion is considered with a recursive method. The narrow-to-broadband conversion is made with a linear regression equation using different coefficients for snow-covered and snow-free pixels [6].

Snow-covered and snow-free observations are also processed separately. A window is classified as snow-covered when the majority of inputs are snow-covered, excluding any existing snow-free input, and vice versa. Thus, the model has the same limitations described for MCD43C3 regarding the use of RossThick-LiSparse-Reciprocal BRDF over snow and regarding the temporal changes of snow reflectance within each composite window. The snow masks used are those embedded in the cloud mask of each input sensor, introducing additional issues due to the lack of consistency between the different masks. Version 1 does not have snow mask information during the AVHRR period, leading to an overestimation of albedo over snow-covered surfaces. AVHRR snow information was added in version 2, but Sanchez-Zapero et al. [38] acknowledged that the magnitude of the snow-covered area in version 2 is still not consistent between sensors, with SPOT/VGT having the largest snow-covered area followed by PROBA-V, NOAA-16/17 and NOAA-7 to 14. The snow-covered area is particularly small during the NOAA-7 to 14 period due to the lack of band 3a in AVHRR-2 sensors.

In this study, we evaluate the two last versions of the multi-sensor product (v1 and v2). The main improvements from v1 to v2 [6] are the addition of snow information during the AVHRR period and the introduction of a VGT-derived BRDF climatology to reduce data gaps and improve sensor consistency. A harmonized pixel identification approach has been also implemented to improve the satellite cross-consistency when dealing with cloud screening and snow detection.

### 2.2. In Situ Observations: FLUXNET and BSRN

FLUXNET is a global confederation of regional networks (EuroFlux, AmeriFlux and AsiaFlux, among others) that use meteorological towers to measure fluxes of $CO_2$, water vapor and energy parameters. The FLUXNET2015 dataset [25], which provides in situ measurements up to 2014 from 212 sites around the globe, was used in this study. Stations register half-hourly up-welling ($SWU$) and down-welling ($SWD$) shortwave irradiance obtained either with an albedometer (e.g., Kipp&Zonen CMP6 or CMP11) or with a paired pyranometer (e.g., Kipp&Zonen CM3, CM11), which have a spectral response within 280–2800 nm. A few stations use silicon pyranometers with a reduced response within 400–1100 nm.

The Baseline Surface Radiation Network (BSRN) [26] is a global network providing high temporal resolution surface broadband radiation measurements since 1992. Data from a selection of BSRN stations are used by the Copernicus Ground-Based Observations for Validation (GBOV), which is a service specifically developed for validating land products such as albedo [39]. BSRN consists currently of 59 active stations plus 18 stations that have contributed in the past but are currently inactive or permanently closed. Out of them, 29 measure both up-welling and down-welling shortwave irradiance with secondary standard pyranometers (250–3000 nm) at 1-min resolution. Compared to FLUXNET, BSRN also measures the direct normal irradiance with a phyreliometer and the diffuse horizontal irradiance with a shaded pyranometer. The sum of diffuse and direct irradiance will be used to quantify the total incoming shortwave irradiance as it is considered unaffected by the possible uncertainties related to the deviation from a perfect cosine response, which might affect any pyranometer measuring the global irradiance component.

From both networks, 41 stations (28 FLUXNET, 13 BSRN) measuring albedo from 2000/02 to 2014/05 at sites with more than 2 snow months per year were selected as candidate stations for the point-to-pixel comparison. All but the three BSRN Antarctic stations are located in the Northern Hemisphere. The albedometer *footprint* in all the stations was calculated as:

$$footprint = 2 \cdot H \cdot tan(HFOV) \tag{1}$$

where $H$ is the effective albedometer height, i.e., the sensor height ($H_{sensor}$) minus the Top-of-Canopy height ($H_{TOC}$), and $HFOV$ is the half field-of-view of the pyranometer. An $HFOV$ of 81° was assumed for all the stations. $H_{TOC}$ was not available at BSRN stations, but all of them are located at sites with low vegetation or no vegetation at all. Therefore, and after visually inspecting the station images available online, an $H_{TOC}$ of 0.25 m was assumed for all BSRN sites.

### 2.3. Spatial Representativeness of the Stations

The direct point-to-pixel comparison is hindered by the different footprints of satellite and in situ sensors [40]. This problem is particularly accentuated in the case of albedo due to its high spatial variability and the small footprint of albedometers, which is limited by the tower height. A spatial representativeness assessment of the stations is therefore required before conducting point-to-pixel comparisons [41]. We make it three steps: (1) assessment of the spatial representativeness with respect to snow cover using a high-resolution snow product, (2) assessment of land cover homogeneity around the station by visual inspecting Sentinel-2 (S2) True Color Images at 10 × 10 m, (3) assessment of the spatial representativeness with respect to albedo using the variogram-based procedure introduced by Román et al. [17] with S2 MSI L2A reflectance at 20 × 20 m. Tests 1 and 2 are made taking into account the exact position of the station with respect to the coarsest pixel validated. Test 3 instead places the station at the center of the pixel validated.

The spatial representativeness with respect to snow is evaluated based on the procedure described in Schwarz et al. [42]. This method uses a high-resolution product to evaluate the variability of a geophysical variable within the coarsest pixel validated.

The high-resolution pixel collocated with the station is compared against the mean of the high-resolution product within the coarse pixel. The method estimates the spatial sampling error (*SSE*), which is the error introduced when estimating a variable over a large area (i.e., the coarse pixel) from a point observation. For this study, the high-resolution product used was NOAA's Interactive Multi-Sensor Snow and Ice Mapping System (IMS) at 1 km, which has provided daily binary snow cover since 2014 [43]. The spatial representativeness was evaluated using all daily IMS 1 km images during 2015, which is the closest year to our study period fully covered by IMS at 1 km. Only one year was used because all snow conditions were covered. For each station, the spatial sampling error was calculated as:

$$SSE = 365 \cdot \frac{1}{N} \sum_{d=1}^{N} |SC_d^{station} - SC_d^{area}| \qquad (2)$$

where $SC_d^{station}$ is the daily snow cover from the IMS 1 km pixel collocated with the station, $SC_d^{area}$ is the average daily snow cover of all IMS 1 km pixels within the larger $0.05 \times 0.05°$, and $N$ is the annual number of daily observations available. *SSE* was multiplied by 365 to analyze the results in terms of annual snow cover duration. Stations with annual $SSE > 5$ days were discarded.

The spatial representativeness with respect to albedo was quantified with the procedure introduced by Román et al. [17], which uses a high-resolution albedo product to derive variogram models in subsets of different sizes around the station. The subsets typically evaluated are $1 \times 1$ km and $1.5 \times 1.5$ km, since this is the spatial domain of MODIS and C3S albedo products. We have also evaluated the $4 \times 4$ km region to assess C3S-AVHRR ($4 \times 4$ km) and GLASS-AVHRR & MCD43C3 ($0.05 \times 0.05°$) grids.

Variograms are derived with Sentinel-2 MSI L2A images. The Bottom-of-Atmosphere (BOA) shortwave reflectance was derived from S2 MSI L2A spectral reflectances using the narrow-to-broadband coefficients proposed by Liu et al. [29]:

$$r_{SW}^{snow-free} = 0.2688 \cdot r_{B02} + 0.0362 \cdot r_{B03} + 0.1501 \cdot r_{B04} + 0.3045 \cdot r_{B8A} + 0.1644 \cdot r_{B11} + 0.0356 \cdot r_{B12} - 0.0049 \qquad (3)$$

$$r_{SW}^{snow-covered} = -0.1992 \cdot r_{B02} + 2.3002 \cdot r_{B03} - 1.9121 \cdot r_{B04} + 0.6715 \cdot r_{B8A} - 2.2728 \cdot r_{B11} + 1.9341 \cdot r_{B12} - 0.0001 \qquad (4)$$

where B02 is centered at 490 nm, B03 at 560 nm, B04 at 665 nm, B8A at 842 nm, B11 at 1610 nm and B12 at 2190 nm.

Variograms were calculated using two Sentinel-2 images per station, during the snow-free and the snow-covered season. In both cases, clear-sky images (cloud cover percentage < 5%) with $SZA < 70°$ were selected. Snow-free images are not available at Neumayer (GVN) and Concordia (DOM) Antarctic stations. No images are available at the South Pole (SPO) observatory, which is outside the S2 coverage. S2 images were processed using the S2 Scene Classification Layer (SCL). Pixels classified as with no data (SCL = 0), defective/saturated pixel (SCL = 1), cloud shadows (SCL = 3) or high cloud probability (SCL = 9) were discarded.

The experimental variograms were derived with the Matheron estimator, a bin width of 20 m, and a maximum lag equal to half the maximum subset distance [17]. A spherical variogram model was fit using the Levenberg–Marquardt algorithm for unconstrained problems. For each subset, the coefficient of variation (*CV*) and the variogram range (*a*), sill (*c*), and nugget (*c*0) [44] were calculated. The range is the distance at which the model flattens. Locations separated by distances closer than the range are spatially correlated. The sill is the constant semi-variance between locations non-autocorrelated (separated by distances equal or larger than the reange). The nugget is the semi-variogram at distance zero and represents the natural variability of the variable assessed.

The variation of variogram properties with the spatial resolution was quantified using the $RAW_{score}$ and $ST_{score}$ defined by Román et al. [17]. $RAW_{score}$ was calculated for the 1–1.5 km and 1–4 km subsets, while $ST_{score}$ was only calculated for the 1–1.5 km subsets because it tends to zero at coarse resolutions due to the high scale index requirement ($R_{SE}$)

(footprint too small compared to subset size). $RAW_{score}$ is independent of the variogram model parameters, so it is the only valid metric when the variogram fit is not good.

### 2.4. Processing In Situ Observations

The point-to-pixel comparison was made with monthly means of blue-sky albedo ($\alpha_{blue}$) at local solar noon. All satellite products evaluated estimate the albedo at local noon.

Instantaneous $SWD$ and $SWU$ measurements were quality controlled using the BSRN tests of extremely rare limits and the consistency test between the radiation components [45]. All values outside those limits were discarded. Time series with a high percentage of points flagged were visually inspected, and if an issue was found, the whole period was removed (e.g., RU-Cok before 2012). Then, the average $SWD_{noon}$ and $SWU_{noon}$ within $\pm 1$ h around local solar noon was calculated if at least 1 FLUXNET 30-min value or 30 BSRN 1-min values were available. In both cases, only clear-sky observations with $SZA < 70°$ were aggregated.

Clear-sky measurements were defined as those having a modified clearness index ($k'_t$) above 0.65 [46,47]:

$$k'_t = \frac{k_t}{1.031 \cdot e^{-1.4/(0.9+9.4/AM)} + 0.1} \tag{5}$$

where $k_t$ is the clearness index ($k_t = SWD/SWD_{TOA}$), and $AM$ is the optical air mass ($AM = 1/cos(SZA)$).

Daily blue-sky albedo was derived as $\alpha_{blue} = SWU_{noon}/SWD_{noon}$. Monthly $\alpha_{blue}$ was then calculated as the average of daily $\alpha_{blue}$ if at least 5 daily values were available. The intra-monthly variability was estimated with the mean absolute deviation of daily measurements.

### 2.5. Processing Satellite Products

The blue-sky albedo of satellite products was calculated from the black and white-sky albedo estimates as:

$$\alpha_{blue} = \alpha_{white} \cdot k_d + \alpha_{black}(1 - k_d) \tag{6}$$

where $k_d$ is the diffuse index. $k_d$ was derived from the diffuse irradiance measurements at BSRN stations and using the Erbs decomposition model [48] at FLUXNET ones:

$$k_d = 1 - 0.099 \cdot k_t \qquad\qquad if(k_t < 0.22) \tag{7}$$
$$k_d = 0.9511 - 0.160 \cdot k_t + 4.3888 \cdot k_t^2 - 16.638 \cdot k_t^3 + 12.336 \cdot k_t^4 \quad if(k_t \geq 0.22)\&(k_t \leq 0.8) \tag{8}$$
$$k_d = 0.165 \qquad\qquad if(k_t > 0.8) \tag{9}$$

In each product, the monthly $\alpha_{blue}$ was calculated by weighting the sub-monthly satellite estimates with the number of days covered by each estimate.

### 2.6. Quality Indicators

The point-to-pixel comparison was made during the period covered simultaneously by GLASS-AVHRR, MCD43C3 v6/6.1 and C3S-SPOT v1/2 (2000/02–2014/05). Some metrics were also calculated during the overlapping period between C3S-AVHRR and C3S-SPOT (2000/02-2005/12) to assess C3S AVHRR and the transition from AVHRR to SPOT/VGT. C3S PROBA was not evaluated due to the lack of FLUXNET data during the PROBA period.

In each station, the absolute and relative mean bias deviation ($MBD$ and $rMBD$), the absolute and relative root mean squared deviation ($RMSD$ and $rRMSD$), and the percentage of monthly estimations within the GCOS uncertainty limits ($max(0.0025, 5\%)$) were calculated. The scatter plots of measured against satellite monthly albedo were also generated for each product and station. Scatter plots include error bars representing the median absolute deviation of daily measurements to evaluate the intra-monthly variability of in situ measurements. Error bars were not included for satellite values due to the low temporal resolution of C3S and GLASS products (3–4 values/months), which hinders the calculation of robust spread metrics. Heat maps of monthly MBD and RMSD per station and product were used to evaluate how the performance of the products changes inter-annually.

The overall product performance was assessed with the boxplots of performance metrics at each station for different snow cover conditions: no snow ($SC = 0$), partly snow-covered ($0 < SC < 1$), and fully snow-covered ($SC = 1$). ERA5 [49] was used to estimate the monthly snow cover conditions at each station. Only stations covered by all the products were included.

## 3. Results

### 3.1. Spatial Representativeness Assessment

The first spatial representativeness test discarded 15 out of the 41 candidate stations due to a heterogeneous snow cover around the station (Figures 1 and S1): 12 FLUXNET stations (AT-Neu, CH-Gru, CN-HaM, k CZ-BK2, DE-Lkb, GL-NuF, IT-MBo, IT-Tor, JP-MBF) and three BSRN stations (BAR, NYA, TIK). The snow cover heterogeneity was caused by the presence of water bodies in the $0.05 \times 0.05°$ pixel or because the station was located in a mountainous region. In pixels containing water bodies, the station overestimates the annual snow cover duration of the $0.05 \times 0.05°$ pixel (CH-Fru, GL-Nuf, SJ-Adv, US-WPT, BAR, NYA, TIK). In the case of mountainous terrain, the station overestimates the pixel snow cover duration if located at the top of (or close to) a mountain (CN-HaM, CZ-BK2, DE-Lkb, IT-MBo, IT-Tor, JP-MBF, US-GBT), while the opposite happens if they are located in a valley (AT-Neu).

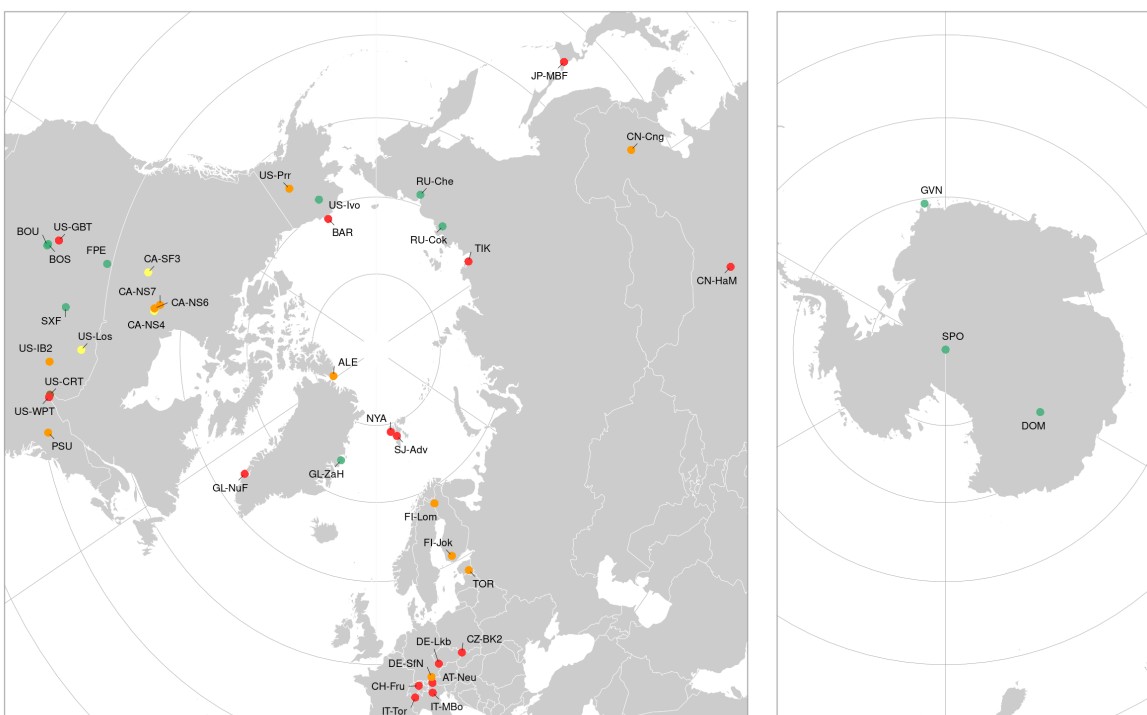

**Figure 1.** Location of the stations used in the study. Stations used in the point-to-pixel comparison are in green. Stations discarded are in red (low snow representativeness), orange (low land cover homogeneity), yellow (low albedo representativeness) and brown (no fully snow-covered measurements). Three letter codes (SSS) indicate BSRN a station. NN-SSS indicates FLUXNET stations.

The second test discarded 11 out of the 26 remaining stations due to a highly heterogeneous land cover inside the $0.05 \times 0.05°$ pixel validated (Figures 1 and S2). These 11 stations were surrounded by urban areas (US-IB2, CN-Cng), heterogeneous mosaics of forests, grasslands and croplands (PSU, FI-Lom, US-Prr, TOR), abrupt topography (ALE), or water bodies (CA-NS6, CA-NS7, DE-SFn).

The 15 remaining stations are summarized in Table 2 and Figure S3. The spatial representativeness of these stations was analyzed quantitatively based on the variograms

derived from S2 snow-covered and snow-free images (Figures S4–S10). The stations could be broadly classified into four main groups, from more to less spatially representative: (1) Antarctic permanently snow-covered stations—GVN, DOM, SPO, (2) high-latitude stations with low vegetation—RU-Cok, RU-Che, US-Ivo, GL-ZaH, (3) stations surrounded by mosaics of croplands and grasslands—SXF, US-CRT, FPE, BOU, BOS, and (4) stations surrounded by a mosaic of forests, natural grasslands and shrub—US-Los, CA-SF3, CA-NS4. Figure 2 shows the variogram analysis of snow-covered Sentinel-2 images of one station in each group. The remaining stations are tabulated in Table 3, and their images are available as supplementary material. The supplementary document also includes the discussion of the variogram models at each station and the evaluation of $RAW_{score}$ and $ST_{score}$ metrics.

**Table 2.** Description of FLUXNET and BSRN stations evaluated with the variogram-based procedure. $H_s$ and $H_{TOC}$ stand for sensor and top-of-canopy heights, respectively. International Geosphere–Biosphere Programme (IGBP) land cover classification: GRA = grasslands, OSH = open shrublands, WET = permanent wetlands, ENF = evergreen needleaaf forests, CRO = croplands, CVM = cropland/natural vegetation mosaics, SNO = snow and ice.

| ID | Name | IGBP | Lat (°) | Lon (°) | Elev (m) | Snow (days) | $H_s/H_{TOC}$(m) | Footprint (m$^2$) |
|---|---|---|---|---|---|---|---|---|
| GL-ZaH | Zackenberg Heath | GRA | 74.47 | −20.55 | 38 | 240 | 3/0.1 | 37 |
| RU-Cok | Chokurdakh | OSH | 70.83 | 147.49 | 48 | 250 | 5/0.5 | 57 |
| RU-Che | Cherski | WET | 68.61 | 161.34 | 6 | 248 | 6/0.5 | 70 |
| US-Ivo | Ivotuk | WET | 68.49 | −155.75 | 568 | 280 | 4/0.3 | 47 |
| CA-NS4 | UCI-1964 burn site wet | ENF | 55.91 | −98.38 | 260 | 176 | 10/2.0 | 101 |
| CA-SF3 | Saskatchewan—W Boreal | OSH | 54.09 | −106.01 | 540 | 146 | 20/18.0 | 25 |
| US-Los | Lost Creek | WET | 46.08 | −89.98 | 480 | 100 | 10.2/2.0 | 104 |
| US-CRT | Curtice Walter-Berger | CRO | 41.63 | −83.35 | 180 | 60 | 2/0.4 | 20 |
| FPE | Fort Peck | GRA | 48.32 | −105.10 | 634 | 85 | 10/0.2 | 123 |
| SXF | Sioux Falls | CRO | 43.73 | −96.62 | 473 | 80 | 10/0.2 | 123 |
| BOS | Boulder | CVM | 40.12 | −105.24 | 1689 | 81 | 10/0.2 | 123 |
| BOU | Boulder | CVM | 40.05 | −105.01 | 1577 | 75 | 300/0.2 | 3785 |
| GVN | Georg von Neumayer | SNO | −70.65 | −8.25 | 42 | 365 | 3/0.0 | 38 |
| DOM | Concordia Dome C | SNO | −75.10 | 123.38 | 3233 | 365 | 3/0.0 | 38 |
| SPO | South Pole | SNO | −89.98 | −24.80 | 2800 | 365 | 3/0.0 | 38 |

**Table 3.** Variogram paramters derived from snow-covered S2 MSI L2A shortwave reflectance images around each station. $CV$ = coefficient of variation, $a$ = range, $c0$ = nugget, $c$ = partial sill.

| | S2 Tile | Date | Side (km) | CV | a | c | c0 |
|---|---|---|---|---|---|---|---|
| GL-ZaH | 27XWC | 07/04/2020 | 1.0 | 0.141 | 868.4 | $8.3 \times 10^{-3}$ | $2.3 \times 10^{-3}$ |
| | | | 1.5 | 0.153 | 385.6 | $9.6 \times 10^{-3}$ | $1.6 \times 10^{-3}$ |
| | | | 4.0 | 0.166 | 487.8 | $9.2 \times 10^{-3}$ | $3.4 \times 10^{-3}$ |
| RU-Cok | 55WEU | 04/04/2020 | 1.0 | 0.069 | 95.0 | $2.1 \times 10^{-3}$ | $4.8 \times 10^{-4}$ |
| | | | 1.5 | 0.063 | 125.9 | $1.9 \times 10^{-3}$ | $4.3 \times 10^{-4}$ |
| | | | 4.0 | 0.055 | 113.6 | $1.5 \times 10^{-3}$ | $3.4 \times 10^{-4}$ |
| RU-Che | 57WWS | 26/04/2020 | 1.0 | 0.093 | 133,845.9 | $4.5 \times 10^{-1}$ | $1.1 \times 10^{-3}$ |
| | | | 1.5 | 0.130 | 1208.2 | $6.1 \times 10^{-3}$ | $2.1 \times 10^{-3}$ |
| | | | 4.0 | 0.132 | 1316.1 | $4.2 \times 10^{-3}$ | $2.8 \times 10^{-3}$ |
| US-Ivo | 04WFB | 20/04/2021 | 1.0 | 0.020 | 197.5 | $1.4 \times 10^{-4}$ | $6.4 \times 10^{-5}$ |
| | | | 1.5 | 0.017 | 231.4 | $9.6 \times 10^{-5}$ | $6.8 \times 10^{-5}$ |
| | | | 4.0 | 0.117 | 919,642.0 | $8.3 \times 10^{-1}$ | $2.9 \times 10^{-3}$ |
| CA-NS4 | 14VNH | 06/05/2020 | 1.0 | 0.394 | 165.6 | $4.2 \times 10^{-3}$ | $2.1 \times 10^{-3}$ |
| | | | 1.5 | 0.387 | 231.0 | $3.9 \times 10^{-3}$ | $2.1 \times 10^{-3}$ |
| | | | 4.0 | 0.463 | 319.9 | $5.6 \times 10^{-3}$ | $3.4 \times 10^{-3}$ |

**Table 3.** *Cont.*

| | S2 Tile | Date | Side (km) | CV | a | c | c0 |
|---|---|---|---|---|---|---|---|
| CA-SF3 | 13UDV | 06/03/2020 | 1.0 | 0.798 | 248.0 | $2.0 \times 10^{-2}$ | $2.7 \times 10^{-3}$ |
| | | | 1.5 | 0.712 | 637.6 | $1.6 \times 10^{-2}$ | $9.2 \times 10^{-3}$ |
| | | | 4.0 | 0.735 | 452.5 | $2.4 \times 10^{-2}$ | $7.0 \times 10^{-3}$ |
| FPE | 13UDP | 20/01/2020 | 1.0 | 0.336 | 339.0 | $1.7 \times 10^{-2}$ | $8.0 \times 10^{-3}$ |
| | | | 1.5 | 0.346 | 313.7 | $1.7 \times 10^{-2}$ | $7.3 \times 10^{-3}$ |
| | | | 4.0 | 0.318 | 498.0 | $1.7 \times 10^{-2}$ | $7.7 \times 10^{-3}$ |
| US-Los | 15TYM | 23/02/2020 | 1.0 | 0.610 | 651.2 | $5.5 \times 10^{-2}$ | $2.9 \times 10^{-3}$ |
| | | | 1.5 | 0.743 | 662.5 | $4.9 \times 10^{-2}$ | $3.0 \times 10^{-3}$ |
| | | | 4.0 | 0.776 | 1248.2 | $3.4 \times 10^{-2}$ | $1.4 \times 10^{-2}$ |
| SXF | 14TPP | 02/01/2021 | 1.0 | 0.296 | 595.7 | $2.9 \times 10^{-2}$ | $3.7 \times 10^{-3}$ |
| | | | 1.5 | 0.319 | 946.3 | $2.4 \times 10^{-2}$ | $8.7 \times 10^{-3}$ |
| | | | 4.0 | 0.261 | 411.7 | $1.8 \times 10^{-2}$ | $6.8 \times 10^{-3}$ |
| US-CRT | 16TGM | 29/02/2020 | 1.0 | 0.202 | 201.6 | $1.1 \times 10^{-2}$ | $3.0 \times 10^{-3}$ |
| | | | 1.5 | 0.278 | 559.1 | $1.5 \times 10^{-2}$ | $9.5 \times 10^{-3}$ |
| | | | 4.0 | 0.286 | 440.5 | $1.8 \times 10^{-2}$ | $7.6 \times 10^{-3}$ |
| BOS | 13TDE | 11/02/2020 | 1.0 | 0.158 | 124.3 | $1.1 \times 10^{-2}$ | $8.0 \times 10^{-4}$ |
| | | | 1.5 | 0.159 | 195.4 | $9.6 \times 10^{-3}$ | $2.1 \times 10^{-3}$ |
| | | | 4.0 | 0.159 | 401.9 | $7.3 \times 10^{-3}$ | $4.2 \times 10^{-3}$ |
| BOU | 13TDE | 08/02/2020 | 1.0 | 0.084 | 807.1 | $3.0 \times 10^{-3}$ | $1.1 \times 10^{-3}$ |
| | | | 1.5 | 0.153 | 306,593.7 | $1.4 \times 10^{-0}$ | $4.5 \times 10^{-3}$ |
| | | | 4.0 | 0.205 | 666.0 | $1.1 \times 10^{-2}$ | $6.4 \times 10^{-3}$ |
| GVN | 29DNB | 04/01/2020 | 1.0 | 0.012 | 85.6 | $3.3 \times 10^{-5}$ | $4.1 \times 10^{-5}$ |
| | | | 1.5 | 0.012 | 90.7 | $3.6 \times 10^{-5}$ | $4.0 \times 10^{-5}$ |
| | | | 4.0 | 0.018 | 85.4 | $1.2 \times 10^{-4}$ | $1.6 \times 10^{-5}$ |
| DOM | 51CWS | 01/01/2021 | 1.0 | 0.009 | 12.5 | $5.3 \times 10^{-4}$ | $4.9 \times 10^{-4}$ |
| | | | 1.5 | 0.010 | 12.5 | $6.1 \times 10^{-4}$ | $5.6 \times 10^{-4}$ |
| | | | 4.0 | 0.018 | 419.1 | $1.4 \times 10^{-4}$ | $6.0 \times 10^{-5}$ |

Based on the varigoram analysis, stations in group 4 (US-Los, CA-NS4, CA-SF3) were discarded for the point-to-pixel comparison. All of them show a high spatial heterogeneity during the snow season due to the large reflectance differences between snow-covered forests and snow-covered grasslands surrounding those forests. US-CRT was also discarded due to the lack of measurements under fully snow-covered conditions ($SC = 1$) during the study period. Therefore, 11 stations were selected as spatially representative enough for the point-to-pixel comparison.

### 3.2. Point-to-Pixel Comparison

The scatter plots of satellite against in situ monthly blue-sky albedo (Figures 3–5) give additional information about the spatial representativeness of the stations. When a similar error appears in all the products at the same station, the insufficient spatial representativeness of the station could be the cause. A common systematic error by all the products was not observed at any station. However, both the random error and the intra-monthly variability increased at the four stations surrounded by mosaics of crops (BOU, BOS, FPE, SXF) during intermediate snow cover conditions ($0 < SC < 1$), which agrees with the spatial representativeness assessment. Random error increases due to reflectance variability caused by the mosaics of crops, their different growing seasons, and the different reflectance of snow over each type of crop. In GLASS and C3S, their low temporal resolution may also contribute to the larger random errors observed. Therefore, results obtained during intermediate snow conditions at these stations should be interpreted with caution. The confidence of the comparison at these stations improves during fully snow-covered conditions with a significant decrease in both the random error and the intra-monthly variability. Regarding the spatial resolution of the products, deviations do not increase systematically from high (C3S SPOT) to low (MCD43C3, GLASS) resolution products, supporting the acceptable spatial representativeness of the stations selected.

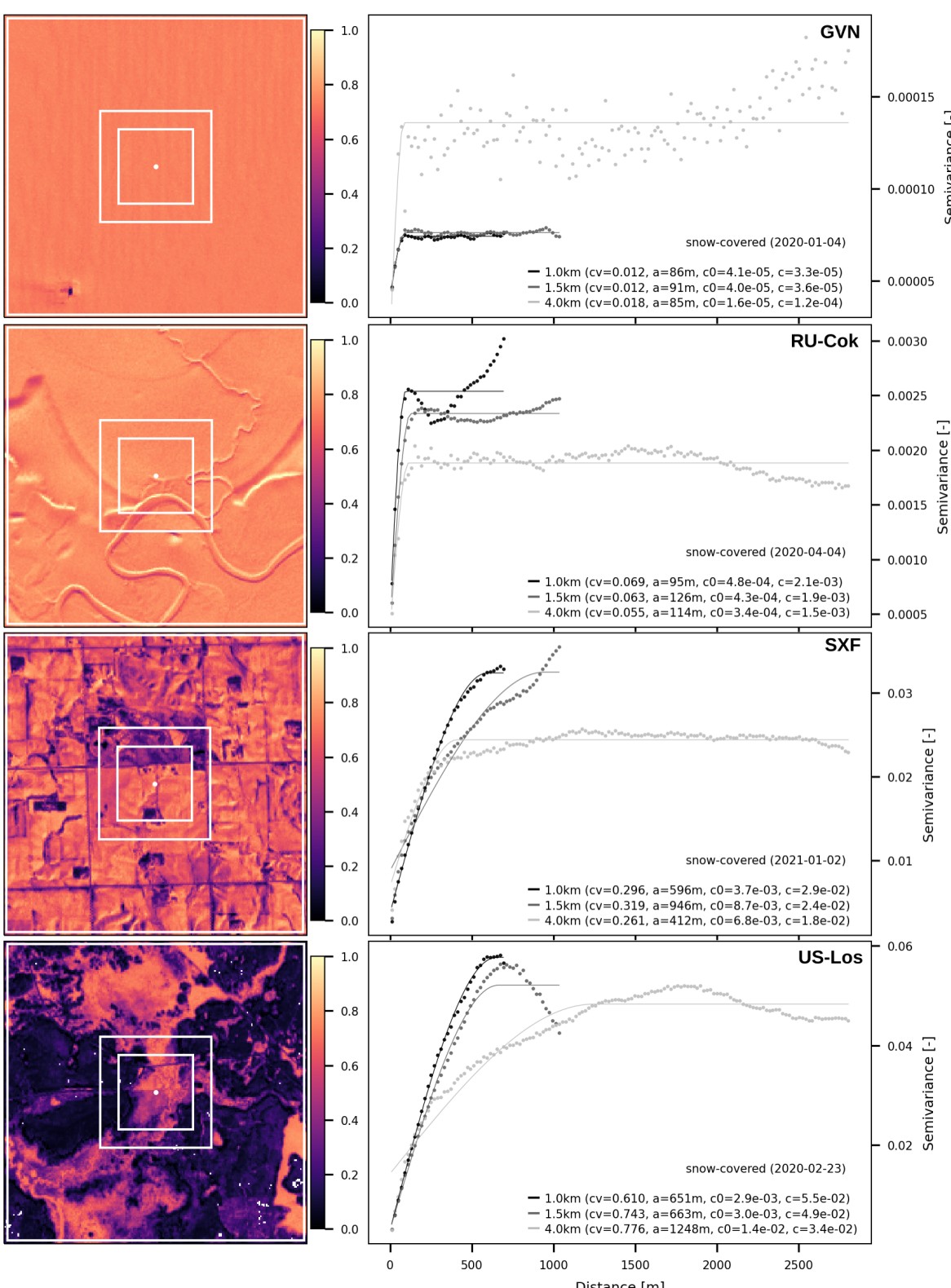

**Figure 2.** Variogram estimator (dots) and spherical models (lines) derived from S2 MSI L2A shortwave reflectance [-] (20 × 20 m) within 4 × 4 km, 1.5 × 1.5 km and 1 × 1 km regions around GVN, RU-CoK, SXF and US-LOS. $CV$ = coefficient of variation, $a$ = range, $c0$ = nugget, $c$ = partial sill.

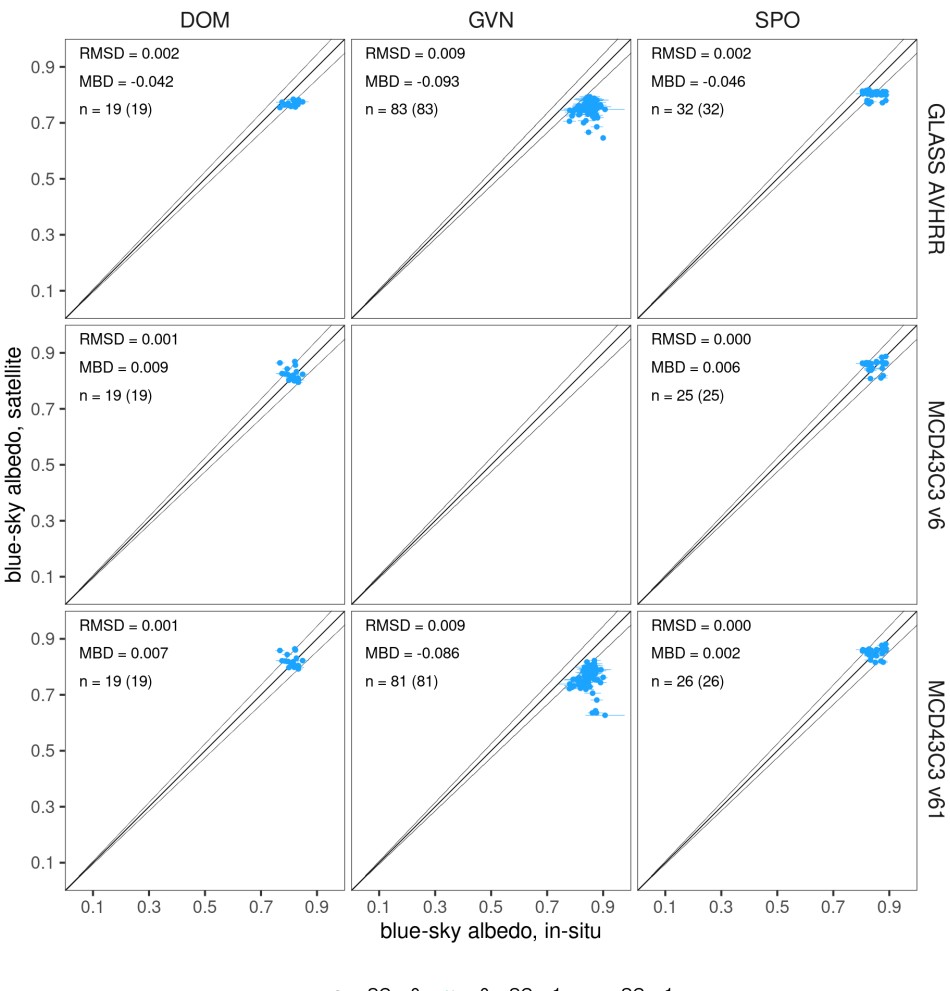

**Figure 3.** Scatter plot of satellite vs in situ monthly blue-sky albedo at BSRN Antarctic stations from 2000/02 to 2014/05. Point color shows monthly snow cover (SC). Error bars represent the mean absolute deviation of daily in situ albedo. Intervals show the GCOS uncertainty requirement of $max(5\%, 0.0025)$. $RMSD$, $MBD$, and $n$ values correspond to $SC = 1$ conditions. The total number of points is shown in brackets. Blank panels correspond to stations not covered by the product.

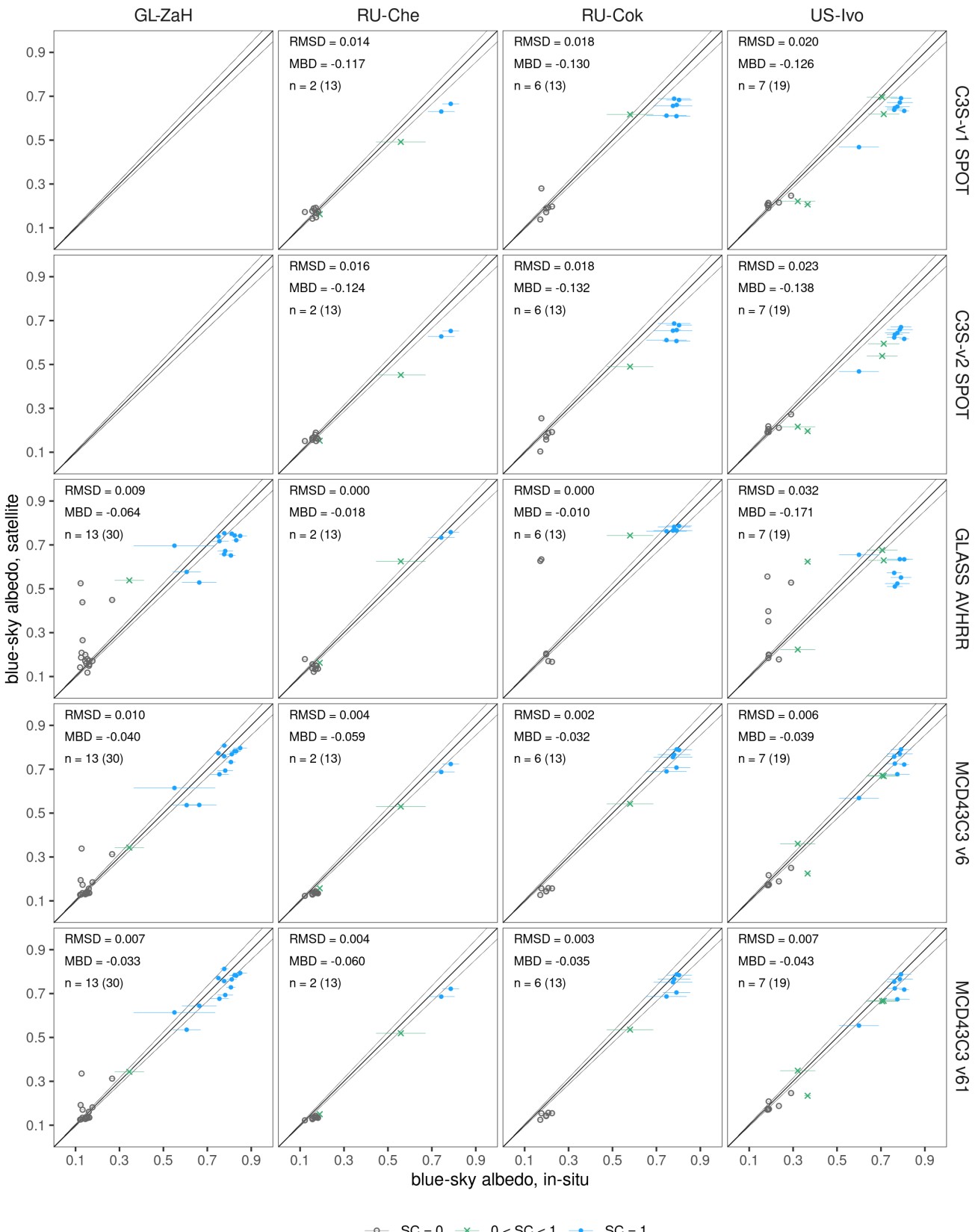

**Figure 4.** Same as Figure 3 but for Fluxnet stations.

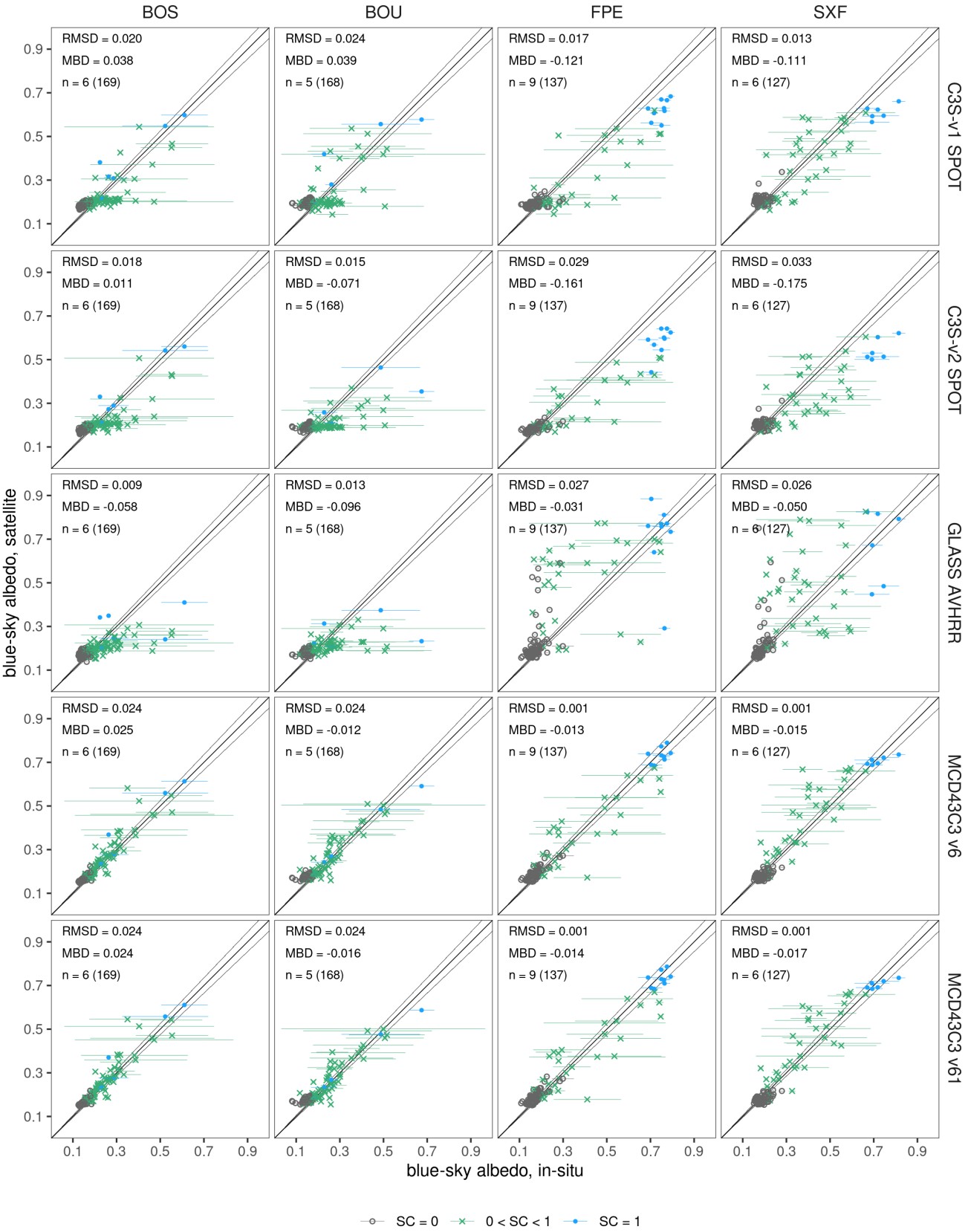

**Figure 5.** Same as Figure 3 but for BSRN stations (excluding Antarctic ones).

MCD43C3 has the best performance over snow, showing both the smallest bias (−0.017) and the smallest RMSE (0.004) in fully snow-covered months. Geographically, MCD43C3 has a small negative bias of around −0.042 at high-latitude stations (RU-Cok, Ru-Che, GL-ZaH, US-Ivo) and unbiased snow albedo at mid-latitude stations surrounded by crops (FPE, SXF, BOU, BOS). MCD43C3 albedo is unbiased at SPO and DOM Antarctic stations, while it shows again a negative bias of −0.086 and GVN (covered only by v6.1). MCD43C3 has the most consistent performance with different snow cover conditions. In all the stations, the sign and magnitude of the bias remain very stable throughout the year (Figure 6) and throughout different snow cover levels (Figure 7). Consequently, MCD43C3 also shows the best performance during the snow-free season, though in this case, C3S-v2 albedo reaches a similar level of accuracy. The underestimation of in situ albedo persists at high-latitude stations during the snow-free months (MBD = −0.031), whereas at mid-latitude stations, MCD43C3 shows a small positive bias of +0.014. The overall RMSD during the snow-free season reduces from 0.004 to 0.001. The largest deviations were obtained at intermediate snow conditions as above mentioned partly due to the worse spatiotemporal representativeness of in situ measurements during this period. MCD43C3 also has the highest percentage of points within the GCOS uncertainty limits at fully snow-covered (66.7%) and partly snow-covered (20.6%) months while reaching similar percentages to C3S-v2 albedo during the snow-free period (11.1 vs. 11.3%). No significant changes are observed between v6.1 and v6 in terms of accuracy. However, v6.1 slightly extends the spatial coverage at high latitudes and low solar elevation conditions.

GLASS-AVHRR performance varies strongly with the snow cover conditions, underestimating albedo during fully snow-covered months and overestimating it during snow-free and partly snow-covered months. The underestimation of albedo during fully snow-covered months by GLASS is larger than that observed in MCD43C3 (−0.050 vs. −0.017), leading also to a larger RMSD (0.013 vs. 0.004) and fewer points within the GCOS uncertainty limit (33.3% vs. 66.7%). The underestimation of snow albedo by GLASS can be seen at all the stations in snow-covered months (Figure 6) and particularly at the three Antarctic stations (MBD from −0.042 to −0.093) (Figure 3). Conversely, GLASS overestimates albedo during both snow-free and partly snow-covered months, and particularly during and after the melting season. In this period, GLASS bias oscillates between +0.10 to +0.25 at all the stations with seasonal snow except for BOU and BOS (21 km far apart) (Figure 6). The period of highly positive bias covers 1 to 3 months: from May to June at high-latitude stations (GL-ZaH, RU-Cok, RU-Che, US-Ivo), covering almost the whole snow-free season, and from February to April at mid-latitude stations (SXF, FPE). In the latter group, GLASS keeps overestimating albedo during the rest of the snow-free season, but the bias reduces to +0.02 to +0.03. Consequently, GLASS has the worst performance during the snow-free season, with the largest MBD (+0.033) and largest RMSD (0.009).

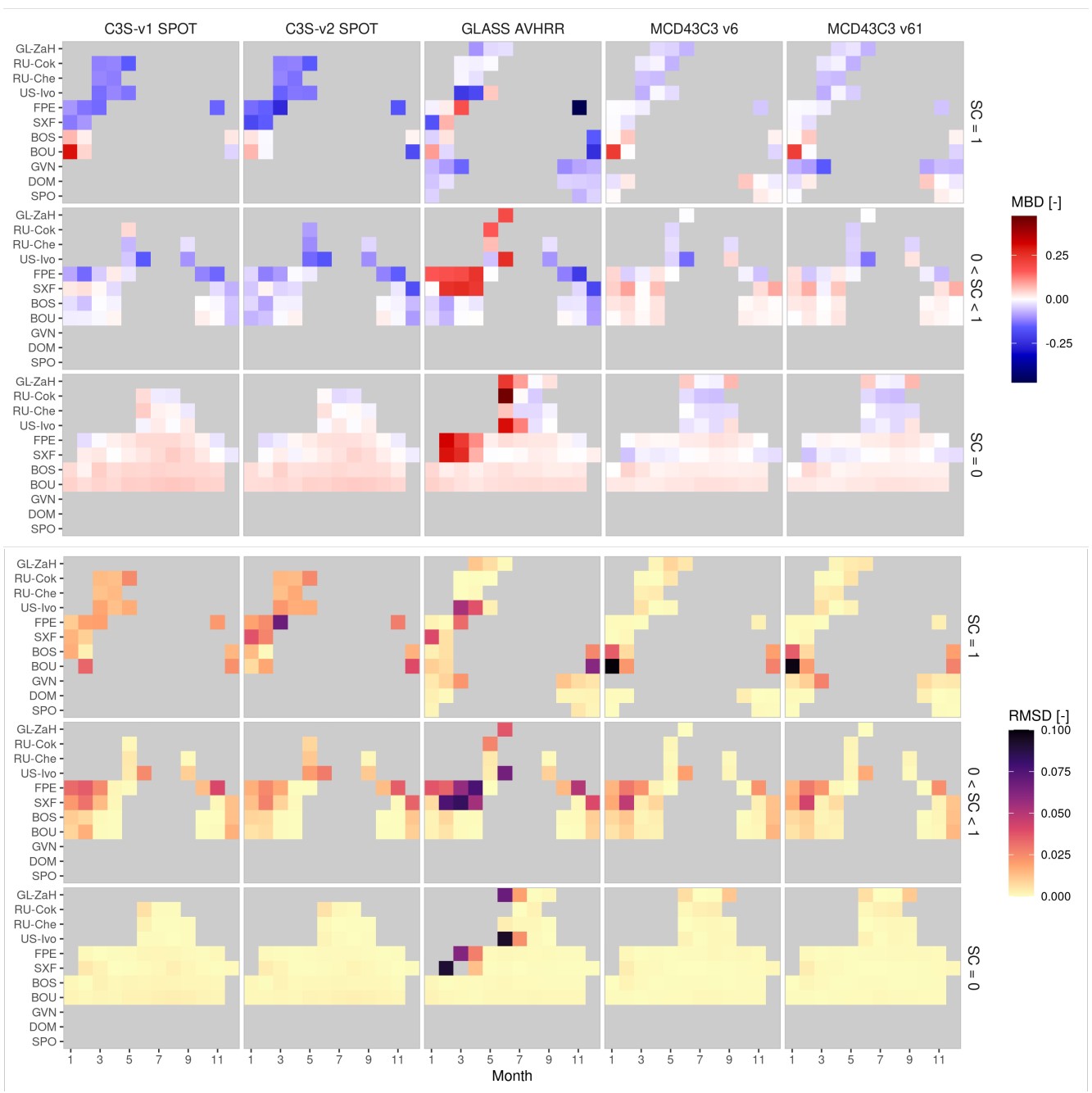

**Figure 6.** Monthly variation of mean bias deviation (MBD) and root mean squared deviation (RMSD) of monthly blue-sky albedo estimations for different snow cover conditions from 2000/02 to 2014/05. Stations are sorted top–down by decreasing latitude.

The bias of C3S albedo changes linearly with the snow cover level, overestimating albedo during snow-free months, slightly underestimating it during partly snow-covered months, and strongly underestimating it during fully snow-covered months (Figure 7). This pattern was observed in the two versions (v1, v2), the two sensors (AVHRR, SPOT/VGT), and at all the stations evaluated (Figure 6). C3S-v2 SPOT has the largest bias overall during the fully snow-covered period (MBD = −0.132) and also the largest RMSD (0.018) and lowest number of points within the GCOS uncertainty limit (median 0%). Its performance improves during the snow-free season when it has a small positive bias of +0.018, an RMSD of 0.001, and 11.3% of points within GCOS uncertainty, which are values similar to those obtained by MCD43C3. When comparing the two versions, C3S-v2 SPOT albedo is some-

how smaller than C3S-v1 albedo by around −0.04 to −0.06 during all seasons, aggravating the albedo underestimation during the snow-covered season.

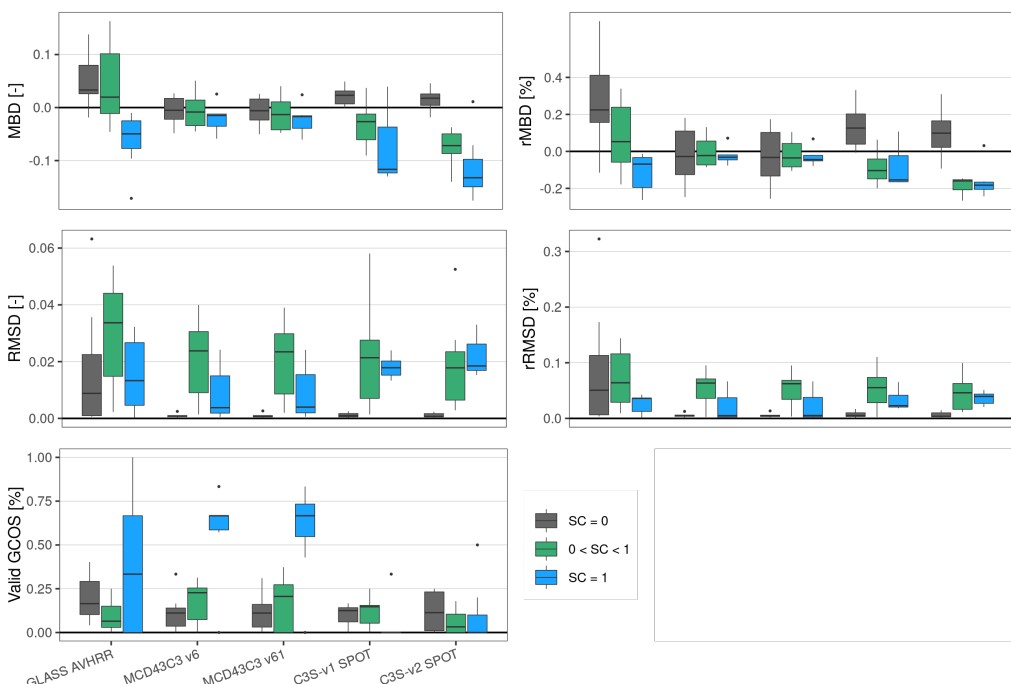

**Figure 7.** Variation of the absolute (MBD) and relative (rMBD) mean bias deviation, absolute (RMSD) and relative (rRMSD) root mean squared deviation, and percentage of points within the GCOS uncertainty requirements (Valid GCOS) with the levels of snow cover (*SC*) level from 2000/02 to 2014/05. Metrics are calculated with the stations covered by all the products (Ru-Cok, Ru-Che, US-Ivo, FPE, SXF, BOU, BOX). Each bopxlot point represents one station.

C3S-AVHRR was evaluated during the overlapping period between AVHRR and SPOT/VGT sensors (Figure A1). A clear improvement is observed from C3S-v1 AVHRR to C3S-v2 AVHRR over snow due to the addition of the snow mask information. The bias during snow-covered conditions reduces from −0.385 to −0.141, bringing AVHRR snow albedo bias close to that of C3S-SPOT (Table A1). However, both periods share the same defects. The bias of C3S-v2 AVHRR also varies linearly with the snow cover level, with a strong negative bias during snow-covered months (MBD = −0.141) and a slight positive bias during snow-free months (+0.010). Moreover, the analysis of the C3S-v2 AVHRR-SPOT overlap period reveals the presence of temporal inconsistencies both during snow-free and snow-covered months.

## 4. Discussion

The differences between products originate from multiple factors ranging from the exact shortwave range of their albedo estimations to differences in their retrieval algorithms: radiance calibration, snow/cloud screening, atmospheric correction, hemispherical integration, and narrow to broadband correction, among others. Products based on BRDF inversion (MCD43C3, C3S albedo) are limited by the capacity of semi-empirical kernels to represent the snow-scattering properties, as they were originally designed to represent snow-free scenarios. Products based on direct estimation (GLASS) fit a unique linear regression model for all types of snow cover situations, which may limit their capability over changing snow cover conditions.

In all products, the blue-sky albedo calculation introduces another source of uncertainty due to the importance of diffuse radiation over high reflective targets and the typically long optical pathway of snow-covered observations. Only clear-sky days with SZA below 70° at local solar noon were used to minimize these effects. Blue-sky albedo

was calculated with the measured diffuse ratio at BSRN stations and with the Erbs model at FLUXNET stations due to the lack of diffuse measurements. We used the BSRN stations to quantify the influence of Erbs models on monthly albedo estimates. Monthly estimates obtained with the Erbs model are around +0.000 to +0.006 higher than those obtained with the measured diffuse index. These values are well below the magnitude of the errors observed and had the same effect on all the products evaluated. However, this suggests that the bias at FLUXNET stations could be slightly more negative than the values reported, particularly in snow-covered situations.

The comparison is also affected by the different spatiotemporal resolution of satellite products and especially by the spatiotemporal mismatch between satellite and in situ measurements. The mismatch error is caused by a sampling error (different co-location of satellite and in situ observations) and a smoothing error (different resolutions). The spatial mismatch error was extensively analyzed in Section 3.1 and in the Supplementary Material, implementing a 3-step methodology to select the stations with the most homogeneous snow cover, land cover and surface reflectance. The results suggest that the spatial representativeness of selected stations is good enough, as differences do not systematically increase either from fine to coarse resolution products or from high to low footprint stations. We refer to Wen et al. [40] for a detailed analysis of spatial heterogeneity in surface reflectance validations.

The temporal sampling error was minimized by calculating the in situ blue-sky albedo around local solar noon ($\pm$30 min), which is the timestamp provided by the three satellite products. The temporal smoothing error originates from the intra-monthly albedo variability and the capacity of both satellite and in situ to gather this variability. The intra-monthly variability is driven by snow cover changes, snow metamorphosis, and vegetation changes, and thus, it is larger during the snowfall and melting season. To mitigate this, we analyzed the product performance in different snow cover conditions. The large variability observed during partial snow covered conditions was significantly reduced during fully snow-covered months (Figures 3–5), making $SC = 1$ results more reliable. Thus, the product analysis is mainly based on the metrics obtained during fully snow-covered months. The ability of each satellite product to gather this variability is limited by the satellite repeat cycle and their snow albedo methodologies. The method applied by MCD43C3 and C3S of processing separately snow and no-snow windows increases their sampling error in partial snow conditions. MCD43C3 compensates for this with a much higher temporal resolution (1 value/day) compared to that of C3S and GLASS (3–4 images/month).

MCD43C3 had the best performance over snow among all the long-term products evaluated (Table 4). It has the smallest bias and RMSD over snow and the most stable bias with different snow cover levels. No significant differences were found between v6.1 and v6 in terms of accuracy. Note that the results could even be probably better if using the highest-resolution MCD43 product available (MCD43A3, 500 m). The good performance of MCD43 over snow was also observed by Wang et al. [22], Stroeve et al. [50] and Song et al. [19], who mentioned the high temporal resolution (1 day) as a key factor in MCD43 capacity to detect snow events. In this study, the only significant issue found over snow was a small negative bias at high-latitude stations and at one of the three Antarctic stations (GVN). Wang et al. [22] and Wang et al. [24] also reported a similar negative bias over snow, suggesting that could be caused by an underestimation by the isotropic diffuse illumination assumption.

GLASS and C3S (v1 and v2) albedo products have significant limitations over snow. Particularly, the bias of both products strongly changes with the snow cover level, overestimating albedo during snow-free months and underestimating it during snow-covered months. GLASS has a systematic negative bias in fully snow-covered conditions (winter in the NH and Antarctic stations) and a positive bias during partly snow-covered and fully snow-covered conditions. This positive bias particularly increases for 1–3 months during and after the melting season at all stations with seasonal snow, leading to a strong positive bias (+0.10 to +0.25) during these months. The most likely cause is that GLASS

snow mask delays the melting season by about 1–3 months, artificially increasing the albedo during this period. GLASS processes pure snow ($r_{490} > 0.4$) and intermediate class B ($0.25 < r_{490} < 0.4$) together, using the same snow/ice BRDF and a unique regression model to estimate broadband albedo from TOA reflectance measurements. The snow melting season may likely fall within the 'intermediate class B', so modeling these intermediate situations as fully snow-covered could be causing the overestimation observed during the melting season. The hypothesis of an insufficient spatial resolution to gather the high spatiotemporal variability of the melting season was discarded because MCD43C3 has the same resolution and it does not show this degradation.

C3S bias changes linearly with the snow cover conditions in both versions (v1, v2) and periods (AVHRR, SPOT/VGT) analyzed. C3S-v2 has the worst performance during the snow season, showing the largest negative bias and RMSD among the products analyzed. C3S-v2 albedo improves during the snow-free season, reaching an accuracy level similar to that of MCD43C3, despite showing a small positive bias. C3S underestimation over snow was also observed by Song et al. [19], Lellouch et al. [51], Sánchez-Zapero et al. [52] and Song et al. [20], who suggested that the low temporal resolution and large compositing window hinder C3S capacity to detect snow events. Our results agree with the previous hypothesis, and indeed, the lower temporal resolution of C3S and GLASS may be one of the reasons of the superior performance of MCD43C3 over snow. The overestimation of snow-free albedo against stations, and also compared to MCD43C3, was also reported by Lellouch et al. [51] and Sánchez-Zapero et al. [52], particularly at regions covered by forests [38]. Our results suggests that this positive bias might exist in all types of land cover, since none of the stations used were surrounded by forests. No improvement was observed during the SPOT/VGT period from v1 to v2. C3S-v2 SPOT albedo is systematically smaller by around $-0.05$ in v2 in all types of snow cover conditions, reducing the overestimation during the snow-free season but aggravating the underestimation during the snow season. The improvements are more noticeable during the AVHRR period. The strong negative bias observed ins C3S-v1 AVHRR has been partly corrected in v2 due to the addition of snow mask information (missing in v1). However, the analysis of the overlapping years between AVHRR and SPOT/VGT reveals that the transition between both sensors is not consistent enough, particularly during the snow season.

**Table 4.** Summary metrics (median MBD and median RMSD) of the different products from 2000/02 to 2014/05 at the stations where all products are simultaneously available (Ru-Cok, Ru-Che, US-Ivo, FPE, SXF, BOU, BOS) and at Antarctic stations (DOM, GVN, SPO).

| Stations | Product | MBD | | | RMSD | | |
|---|---|---|---|---|---|---|---|
| | | $SC = 0$ | $0 < SC < 1$ | $SC = 1$ | $SC = 0$ | $0 < SC < 1$ | $SC = 1$ |
| Ru-Cok, Ru-Che, US-ivo, FPE SXF, BOU, BOS | GLASS-AVHRR | 0.033 | 0.020 | $-0.050$ | 0.009 | 0.034 | 0.013 |
| | MCD43C3 v6 | $-0.005$ | $-0.008$ | $-0.015$ | 0.001 | 0.024 | 0.004 |
| | MCD43C3 v61 | $-0.006$ | $-0.013$ | $-0.017$ | 0.001 | 0.023 | 0.004 |
| | C3S-v1 SPOT | 0.023 | $-0.027$ | $-0.117$ | 0.001 | 0.021 | 0.018 |
| | C3S-v2 SPOT | 0.018 | $-0.072$ | $-0.132$ | 0.001 | 0.018 | 0.018 |
| DOM, GVN, SPO | GLASS-AVHRR | - | - | $-0.046$ | - | - | 0.002 |
| | MCD43C3 v61 | - | - | 0.002 | - | - | 0.001 |

## 5. Conclusions

We have evaluated the performance of long-term (+20 years) albedo products over snow, using 11 FLUXNET and BSRN spatially representative stations as reference. The stations were selected with a three-step procedure that uses high-resolution snow (IMS 1 km) and optical (Sentinel-2, 20 m) satellite products to assess the spatial variability of snow and albedo around the stations.

MCD43C3 is the most reliable product to monitor snow albedo, showing the smallest bias and RMSE over snow, and the most consistent performance with different snow cover conditions. The two versions evaluated, 6 and 6.1, have similar performance, with v6.1

just increasing slightly the coverage at high latitudes. On the contrary, the quality of both GLASS-AVHRR and C3S-v1 and v2 degrades over snow. The bias of both products strongly changes with the snow cover conditions, underestimating albedo over snow and overestimating it over snow-free surfaces. The bias of GLASS-AVHRR increases particularly during the melting season, which is most likely due to an artificially extended snow season. The C3S multi-sensor albedo has improved from v1 to v2, particularly during the AVHRR period. However, both C3S-v2 AVHRR and SPOT have the largest negative bias overall during the snow season, and moreover, temporal inconsistencies were observed in the transition periods between both sensors.

**Supplementary Materials:** The following supporting information can be downloaded at: https://www.mdpi.com/article/10.3390/rs14153745/s1, Figure S1: Sentinel-2 MSI L2A True Color Image (TCI) (10 × 10 m) in the 0.05 × 0.05 ° pixels containing stations discarded due to an heterogeneous snow cover. The white cross represents the station location. The red rectangle shows the 0.05 × 0.05 ° pixel.; Figure S2: Same as Figure S1 but for stations discarded due to an heterogeneous land cover; Figure S3: Same as Figure S1 but for stations with homogeneous land and snow cover; Figure S4: Variogram estimator (dots) and spherical models (lines) derived from S2 MSI L2A shortwave reflectance [-] (20 × 20 m) within 4 × 4 km, 1.5 × 1.5 km and 1 × 1 km regions around the station: (a) DOM—Concordia Station, Dome C, (b) GVN—Georg von Neumayer (c) RU-Cok—Chokurdahk. $CV$ = coefficient of variation, $a$ = range, $c0$ = nugget, $c$ = partial sill; Figure S5: Same as Figure S4 but for:(a) BOS—Boulder and (b) US-Los—Lost Creek; Figure S6: Same as Figure S4 but for (a) CA-NS4—UXI-1964 burn site and (b) CA-SF3—Saskatchewan W Boreal; Figure S7: Same as Figure S4 but for (a) BOU—Bolder and (b) FPE—Fort Peck; Figure S8: Same as Figure S4 but for (a) SXF—Sioux Falls and (b) US-CRT—Curtice Walter-Berger cropland; Figure S9: Same as Figure S4 but for (a) RU-Che—Cherski and (b) US-Ivo—Ivotuk; Figure S10: Same as Figure S4 but for GL-ZaH; Table S1: Quantitative assessment of the spatial representativeness of the stations with respect to albedo based on the coefficient of variation ($CV$), $ST_{score}$, and $RAW_{score}$ derived from variograms at 4 × 4 km, 1.5 × 1.5 km and 1 × 1 km.

**Author Contributions:** Conceptualization, R.U.; methodology, R.U. and C.L.; validation, R.U. and C.L.; formal analysis, R.U.; investigation, R.U., C.L. and F.C.; data curation, R.U. and F.C.; writing—original draft preparation, R.U.; writing—review and editing, C.L., F.C. and N.G.; visualization, R.U.; supervision, N.G.; project administration, N.G. All authors have read and agreed to the published version of the manuscript.

**Funding:** This research received no external funding.

**Data Availability Statement:** All products are freely available online: C3S v1 and v2 surface albedo and ERA5 snow depth at the Climate Data Store; MCD43C3 v6 and v61, MCD12C1 at NASA's Level-1 and Atmosphere Archive and Distribution System Distributed Active Archive; NOAA's IMS snow cover at National Snow and Icde Data Center; and GLASS-AVHRR at www.glass.umd.edu/. Sentinel-2 images were retrieved from ESA's Copernicus Open Access Hub.

**Acknowledgments:** The support provided by DG DEFIS, i.e., the European Commission Directorate General for Internal Market, Industry, Entrepreneurship and SMEs, and Copernicus Programme is gratefully acknowledged. We acknowledge all the satellite data providers as well as FLUXNET and BSRN for maintaining and providing the ground stations used in the study.

**Conflicts of Interest:** The authors declare no conflict of interest.

## Appendix A. Evaluation of C3S Abledo during the Overlap Period between SPOT/VGT and AVHRR

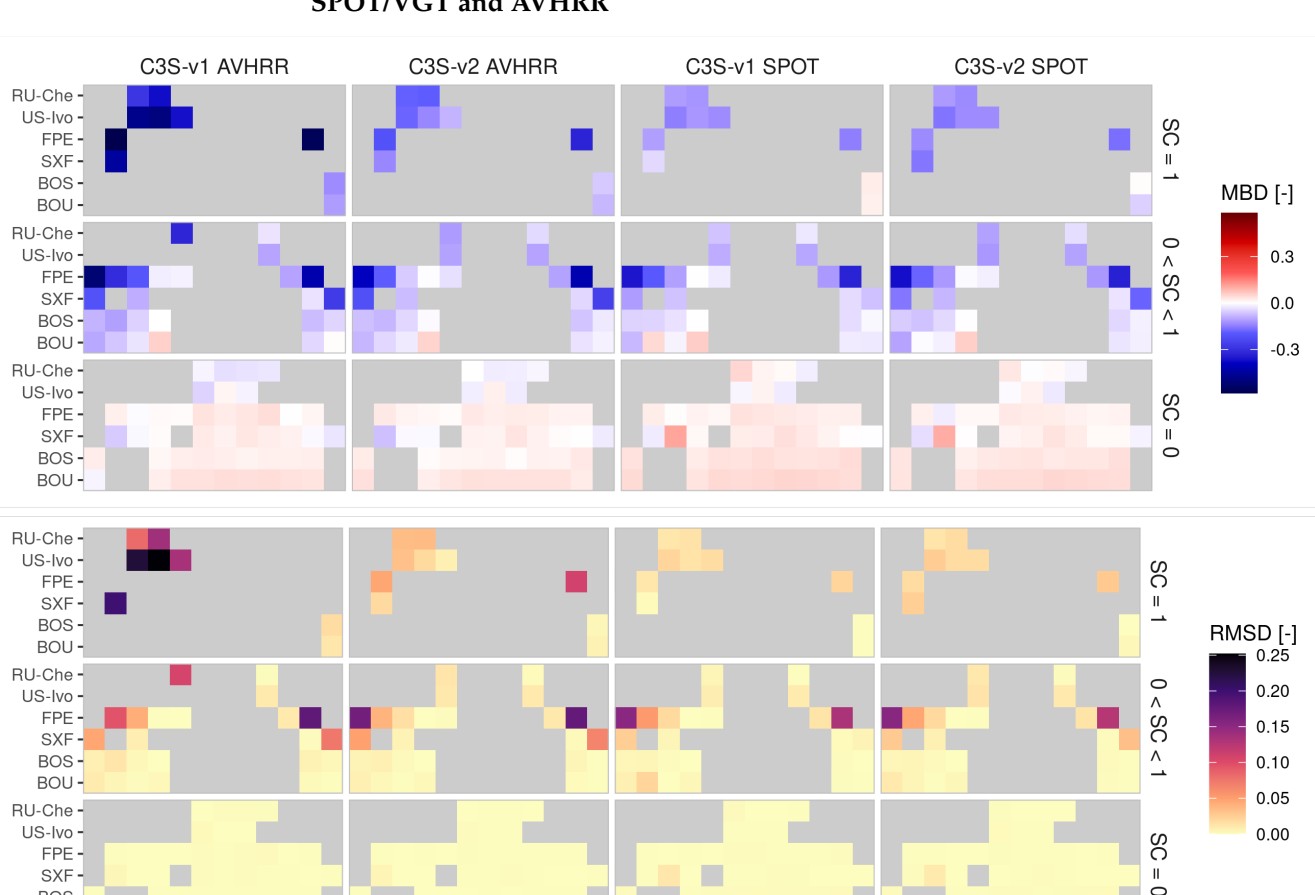

**Figure A1.** Monthly variation of mean bias deviation (MBD) and root mean squared deviation (RMSD) of monthly blue-sky albedo estimations for the different snow cover conditions from 2000/02 to 2005/12, i.e., overlap period between C3S-AVHRR and C3S-SPOT. Stations are sorted top-down by decreasing latitude.

**Table A1.** Summary metrics (median MBD and median RMSD) of products from 2000/02 to 2005/12 at the stations where all products are simultaneously available during that period (Ru-Che, US-Ivo, FPE, SXF, BOU, BOS).

| Stations | Product | MBD | | | RMSD | | |
|---|---|---|---|---|---|---|---|
| | | $SC = 0$ | $0 < SC < 1$ | $SC = 1$ | $SC = 0$ | $0 < SC < 1$ | $SC = 1$ |
| Ru-Che, US-ivo, FPE, SXF, BOS, BOS | C3S-v1 AVHRR | 0.015 | −0.132 | −0.385 | 0.001 | 0.019 | 0.152 |
| | C3S-v2 AVHRR | 0.010 | −0.091 | −0.141 | 0.000 | 0.027 | 0.021 |
| | C3S-v1 SPOT | 0.024 | −0.060 | −0.080 | 0.001 | 0.024 | 0.008 |
| | C3S-v2 SPOT | 0.017 | −0.092 | −0.134 | 0.001 | 0.023 | 0.019 |

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
