# Peer review of "Comparison of Long-Term Albedo Products against Spatially Representative Stations over Snow"

_remotesensing, doi:10.3390/rs14153745_

Round 1

Reviewer 1 Report

This manuscript has made a unique comparison among different satellite based albedo products against the ground network observations. The work may involve a large amount of data processing and the results should be of statistic meaningful, and be of some interest and helpful to surface radiation retrieval studies.  In general, this manuscript is clearly written and well organized. However, there are still some concerns regarding focus of the content and format of results displaying.

Major concerns:

(1) As indicated by the title, this manuscript should focus on comparison of albedo for full snow or partial snow covering cases i.e. SC=1 or 0<SC<1 in the text. The snow-free SC=0 cases should be excluded and they are not necessarily shown in Figures 2 – 8. Not sure if the calculation in Fig.7 includes all cases, if so, probably need to re-calculate only including snow cases. While the authors can still briefly describe the scenario of snow-free cases in the text just for some readers’ curiosity.

(2) The authors have put too much on describing, displaying and discussing spatial representativeness. That makes Sections 2.3 and 3.1 too lengthy. The authors even add 8 extra figures in Appendix A to further showing assessment of the spatial representativeness. I strongly suggest a significantly concise work (text and figures reducing) to these sections. Also, remain scenario for only one or two stations as example in Appendix A.

(3) In Section 2.1 where the different satellite products are introduced, the authors may add some content indicating what’s the difference for these products in deriving snow albedo, for example, what BRDF models used? how to deal with partial snow? etc. As it is, the authors could add one more column in Table 1 describing what albedo retrieval algorithm/brdf model etc.

Other comments by Line number:

L22: typo, “due to is key” should be “due to its key”?

L29: “IPCC (Intergovernmental Panel On Climate Change)”

L36: “SPOT” better be “SPOT-VGT” more exactly; typo, PROVA-V should be PROBA-V

L37: typo, Terra-Acqua should be Terra-Aqua, same typo in Table 1

L52: “the low footprint of the sensors”, what the sensors here refer to? If they refer to satellites, the footprint should be “large” instead of “low”, a little confusing here.

L109: “Visible 300-400 nm”? is it “300-700 nm”?

L138: Normally the RossThick-LiSparse-Reciprocal BRDF model is designed mainly for vegetated land and is not suitable for snow/ice, there are references discussing some more specific BRDF model for snow/ice. The author may make a note here. This is also corresponding to Major concern (3).

Figures 2-3: Correlated to (2) above, showing a couple of stations as example, tabulating the rest stations.

Figures 4-6: It’s really hard to discriminate different dots (too small and mixed). Try to use different symbols for different SC conditions. Fig.5-6 should share the same caption as Fig.4, i.e. "same as Figure 4 but for...."

Section 3.2: There are a lot of data values/numbers embedded in text for comparison of different products. It should be more straightforward to put them in a table and make concise text.

Reviewer 2 Report

I suggest to publish this paper as it stands.

Reviewer 3 Report

This paper attempts to inter-compare satellite-based snow products. Such initiative is very challenging and despite a huge work achieved, after reading the submitted paper I'm not convinced about the usefulness of the outcomes. Actually, there exists so many discrepancies in the way to generate a snow albedo product between sensors that I'm not even convinced it is feasible. Of course, establishing diagnostic is quite useful but each station is a case studty and extrapolation or generality of the results is quite difficult. Mostly, upscaling issue is regarded in this paper (in the body of the text and in the long Appendix also). But footprint size is not the only issue. The biggest issue is that some satellite processing chains mix snow-free and snow observations whereas some others apply the majority rule (snow or not snow). In fact, it would be necessary to go back to previous levels of data processing for a fair comparison and keep track of the original observations and quality flag. Other issue: diffuse radiation is important over high reflective targets such as snow with grazing illumination. Estimate of blue sky albedo could be different only due to an inappropriate description of scattering effects within the atmosphere and the long optical pathway. Other causes of potential differences amongst many: clumped vegetation and shaded snow vs sunlit snow (different orbit pass times between satellites mean big differences because snow occurence is coherent with large solar zenith angles and large proportion of shadows on snow with large variations over short time), topography (slopes and aspects resolution scales), melting periods with different snow metamorphism at different times of the day (as compared satellites observe at different times), narrowband to broadband conversion, etc. If all cimulative errors were accounted for, the conclusions would be very limited.

For the reasons aforementioned, my overall opinion is a rejection of the paper. Furthermore, since I do not see how this work could be  improved, I do not encourage the authors to resubmit.

Round 2

Reviewer 1 Report

The revised version has been significantly improved. It generally satisfies the requirements for publishing as it is.  

Author Response

Thanks for the positive comment.

Reviewer 3 Report

I would like to thank the authors to take into account my remarks and suggestions in the modifications of their manuscript.

Author Response

Thanks for the positive comment.